# Hygiene Defense Behaviors Used by a Fungus-Growing Ant Depend on the Fungal Pathogen Stages

**DOI:** 10.3390/insects10050130

**Published:** 2019-05-05

**Authors:** Ernesto Bonadies, William T. Wcislo, Dumas Gálvez, William O.H. Hughes, Hermógenes Fernández-Marín

**Affiliations:** 1Centro de Biodiversidad y Descubrimiento de Drogas, Instituto de Investigaciones Científicas y Servicios de Alta Tecnología (INDICASAT AIP), Apartado 0843-01103, Panama, Republic of Panama; 2Programa de Maestría en Entomología, Vicerrectoría de Investigación y Postgrado, Estafeta Universitaria 0824, Universidad de Panama, Republic of Panama; 3Smithsonian Tropical Research Institute, Apartado Postal 0843-03092, Panama, Republic of Panama; 4School of Life Sciences, University of Sussex, Brighton BN1 9QG, UK

**Keywords:** parasites, Attini, *Trachymyrmex*, immune system, social insects, *Metarhizium*, *Escovopsis*

## Abstract

Parasites and their hosts use different strategies to overcome the defenses of the other, often resulting in an evolutionary arms race. Limited animal studies have explored the differential responses of hosts when challenged by differential parasite loads and different developmental stages of a parasite. The fungus-growing ant *Trachymyrmex* sp. 10 employs three different hygienic strategies to control fungal pathogens: Grooming the antibiotic-producing metapleural glands (MGs) and planting or weeding their mutualistic fungal crop. By inoculating *Trachymyrmex* colonies with different parasite concentrations (*Metarhizium*) or stages (germinated conidia or ungermianted conidia of *Metarhizium* and *Escovopsis*), we tested whether ants modulate and change hygienic strategies depending on the nature of the parasite challenge. There was no effect of the concentration of parasite on the frequencies of the defensive behaviors, indicating that the ants did not change defensive strategy according to the level of threat. However, when challenged with conidia of *Escovopsis* sp. and *Metarhizium brunneum* that were germinated or not-germinated, the ants adjusted their thygienic behavior to fungal planting and MG grooming behaviors using strategies depending on the conidia germination status. Our study suggests that fungus-growing ants can adjust the use of hygienic strategies based on the nature of the parasites.

## 1. Introduction

Living in societies may increase the transmission of parasites within groups and this can be exacerbated by high genetic relatedness among group members [1,2]. Natural selection on social organisms has driven the evolution of strategies for disease prevention and control, including rapid detection of pathogens, increased immune responses, waste and corpse disposal, social grooming, prophylactic medication, and self-medication [3,4,5,6,7]. The availability of different hygienic strategies could enable individuals or groups to deploy them differentially, according to the nature of the parasite threat and the efficiency of each defense strategy [8]. However, little is known about how individuals determine which particular hygienic strategies to deploy or what triggers the switch between them [9].

Fungus-growing ants (*Formicidae*: *Myrmicinae*) such as *Trachymyrmex* (Forel) form an obligatory mutualism with a fungi crop (*Basidiomycota*: *Lepiotaceae* and *Pterulaceae* [10]), and both ant and crop are challenged by diverse parasites [11,12]. These ants use several different behavioral strategies to deal with parasites [13,14], relying primarily on grooming with their antibiotic-producing metapleural gland (MG), or planting or weeding the fungal mutualist [15]. Some species also utilize antibiotic-producing actinomycete bacteria, while others, such as *Trachymyrmex* sp. 10 lack this symbiont and instead make more use of MG grooming or fungal planting [15]. Here, we used *Trachymyrmex* sp. 10 to test the hypothesis that hosts will alter their behavioral defense strategy either quantitatively or qualitatively, depending on the dose or nature of the parasite challenge.

## 2. Materials and Methods

We collected colonies (ants and mutualist fungus gardens) of *Trachymyrmex* sp. 10 on the campus of the University of Panama in Panama City, Panama. Colonies were kept in plastic boxes under laboratory conditions at the Smithsonian Tropical Research Institute in Gamboa. The colonies had wet paper towels placed in the boxes to maintain humidity and were fed twice a week with corn flour, oatmeal, and occasionally leaves of *Lagerstroemia speciosa*. We used two parasites in our experiments: (i) The entomopathogenic fungus, *Metarhizium brunneum* (Petch) and (ii) a pathogen of the mutualist fungus garden, *Escovopsis* sp. (Muchovej & Della Lucia). We isolated both parasites from a local study site in Gamboa and cultured them on agar plates. Isolates of both parasites had >99% conidia viability when used in the experiments. Voucher specimens from the study were deposited in the collections of the Museo de Invertebrado G.B. Fairchild, Universidad de Panamá and in the Smithsonian Tropical Research Institute.

### 2.1. Experimental Groups and Study Behaviors

From each collected colony, we created groups of 20 workers, randomly chosen and kept in Petri dishes with wet paper strips and 0.5 g of the fungus garden placed in the center of the dish. We quantified four different behaviors to determine the response of ants to pathogens, following Fernández-Marín et al. [15]: (i) Fungal grooming with ants licking the surface of the mutualist fungus to remove contaminated particles which are stored and treated in the infrabuccal pocket and later discarded as pellets; (ii) metapleural gland (MG) grooming, with ants rubbing their forelegs over the MG opening to transfer antibiotic MG secretions to a site of contamination; (iii) fungal cultivar planting, with ants placing healthy pieces of the fungus garden on a contaminated area of the garden; and (iv) weeding, with workers removing a piece of the garden that is then placed in a garbage dump.

### 2.2. Hygienic Responses to Different *Metarhizium brunneum* Pathogen Concentrations

We performed inoculations with conidia of the entomopathogenic fungus *Metarhizium brunneum*. We tested three different concentrations, high (~2.3 × 10^8^ conidia), medium (~6.0 × 10^7^ conidia), and low (~2.1 × 10^7^ conidia). We inoculated the conidia by placing them on a piece of parafilm, which was then rubbed on the surface of the fungal gardens (methods from [12]). The high, medium, and low concentrations correspond to areas in the petri dish of 25 mm^2^, 9 mm^2^, and 4 mm^2^, respectively. In this experiment, we used 30 colonies, and from each colony we formed one experimental group with 20 workers and 0.5 g of the fungus garden. We used 10 experimental groups for each pathogen concentration. The colonies used for each concentration were selected randomly.

### 2.3. Hygienic Responses to Different Stages of the Pathogen

For this experiment, we performed inoculations using an entomopathogen fungus, *M. brunneum*, and a pathogen of the fungal cultivar, *Escovopsis* sp. For both pathogens, we investigated the effect of the developmental stage of the pathogen by using non-germinated and germinated conidia. We germinated 4 mm^2^ of conidia on a piece of fungus garden for 30 h to obtain germinated conidia. As a control for conidia, we used 4 mm^2^ of talcum powder on the fungus garden. As a control for germinated conidia, we inoculated 4 mm^2^ of talcum powder on a piece of fungal garden and left it for 30 h. In this experiment, we used 60 colonies, and from each colony we formed one experimental group with 20 workers and 0.5 g of the fungus garden. We used 10 experimental groups for each stage per pathogen and control. The colonies used for each concentration were selected randomly. We applied the germinated conidia, ungerminated conidia, or control treatments by gently rubbing it on to the fungus garden in the experimental groups. We used 10 experimental groups for each treatment (*n* = 60). 

### 2.4. Statistical Analyses

For the analyzes, we recorded the total number of times that each behavior was observed during one hour, and this was divided by the average number of workers attending the fungus garden at the beginning of each 10 min period during one hour (following methods used in [12]). The observations were conducted with the aid of a 70× stereomicroscope. Data from both experiments were analyzed with PERMANOVA [16]. For the concentration experiment, we explored the possibility of collinearity between behavioral responses for the experiment with different pathogen concentrations by using Pearson correlations. For the stages experiment, in addition to PERMANOVA, we used one-way permutation tests as a post-hoc analysis for pairwise comparisons. 

## 3. Results

Overall, the hygienic responses of workers did not vary with parasite concentration (PERMANOVA: F_2,29_ = 1.0, *p* = 0.4). We did not detect significant correlations between MG grooming and weeding or planting (respectively, r = −0.19, t = −1.01, df = 28, *p* = 0.32; r = −0.34, t = −1.94, df = 28, *p* = 0.06), nor between planting and weeding (r = 0.15, t = 0.80, df = 28, *p* = 0.43; Figure 1).

The developmental stage of the parasite triggered different responses by workers (PERMANOVA: F_1,39_ = 13.4, *p* < 0.0001). Against both parasites, workers used MG grooming more frequently than planting when they encountered germinated conidia of the pathogens (one-way permutation test: *Escovopsis*, Z = −2.3, *p* = 0.006, *Metahizium*, Z = −2.5, *p* = 0.008), and planted more frequently than MG grooming when they encountered non-germinated conidia (*Escovopsis*, Z = 2.3, *p* = 0.006, *Metarhizium*, Z = 2.3, *p* = 0.04; Figure 2). There were no differences in frequencies of behaviors in control treatments when they simulated germinated or non-germinated conidia (one-way permutation test: MG grooming, Z = 0.51, *p* > 0.05, planting, Z = −0.46, *p* > 0.05; Figure 2). There was also no effect of the parasite on the hygienic behaviors when comparing *Metarhizium* and *Escovopsis* (F_1,39_ = 2.7, *p* = 0.06; Figure 2).

## 4. Discussion

Fungus-growing ants are able to detect the presence of parasitic fungi and have previously been shown to increase cleaning rates depending on the parasite concentration [12]. In this study, we found that the ants adjust which behavior is used as the main hygienic strategy depending on the nature of the parasite threat, using primarily the metapleural gland (MG) grooming against germinated conidia from both *Metarhizium* and *Escovopsis* fungi, and plant their mutualistic fungus against ungerminated conidia. Moreover, the ants did not adjust their behavior based on the conidia concentrations tested. However, our results showed that increasing *M. brunneum* conidia concentration increased the hygiene behaviors. Previous studies reported a positive association between an increase on the pathogen concentration and use of the hygiene defense [12].

*Trachymyrmex* sp. 10 lack visible antibiotic-producing bacteria on their cuticle, and the MG secretions are consequently the main source of antimicrobial compounds to fight parasites, as is also the case for *Atta* leafcutter ants [17,18], and derived *Attine* ants in general employ the MG more than lower *Attine* ants [19]. The synthesis and deployment of MG-grooming secretions entails a high metabolic cost, approximately 13–20% of basal metabolism [20], so adjusting its use according to the threat will be beneficial. However, the energetic cost of planting behavior is not known, but is likely significant as the ants have to collect substrate for, and clean, the fungal mutualist in order to grow it. More studies are needed to understand planting costs and function [15]. 

Another reason why ants could change between behaviors may be related to the specificity, efficacy of the hygienic behavior, or immunity system. The fungal mutualist produces a wide variety of compounds with antimicrobial activity [21], and there is evidence that they inhibit endophytes [22] and mycoparasites, such as *Escovopsis* sp. [23], and also slow down *Metarhizium* growth [24,25]. Additionally, planting behavior covers the pathogenic conidia that may be beneficial. For example, *Metarhizium* sp. requires oxygen during germination [26], so planting behavior could potentially create a microenvironment unfavorable to the germination or growth of the pathogen [27]. Fungal symbionts in *Acromyrmex* ants can inhibit *Metarhizium* fungal germination and growth [25], but both *Metarhizium* spores and hyphae are sensitive to MG compounds, as are *Escovopsis* spores, but not *Escovopsis* hyphae [8]. On other hand, *Escovopsis* germination and growth is sensitive to MG compounds, but not *Metharizium* [15]. Moreover, social immune strategies could modulate the grooming behavior with each stage of the infection, as in termites [28]. Those studies suggest that non-germinated conidia are susceptible to fungal planting strategies, while germinated conidia could to be susceptible to MG compounds. Using the same compound to also control a parasite could lead to the faster evolution of resistance, as has been abundantly illustrated by human use of pesticides and antibiotics [29]. Using different compounds and hygienic strategies with different modes of action has been proposed to avoid pesticide resistance in agricultural ecosystems and antibiotic resistance in medicine [29]. Fungus-growing ants arose ~50–60 million years ago, and utilizing different strategies according to the nature of the parasite threat may have helped them to avoid resistance to their antimicrobial compounds.

## 5. Conclusions

The fungus-growing ant *Trachymyrmex* sp.10 employs a diversity of mechanisms to avoid and control parasitic fungi infection. Following our results, these ants use primarily their metapleural gland secretions against germinated conidia, while employing the fungal planting behavior when challenged by ungerminated conidia. Our study suggests that fungus-growing ants can adjust the use of hygienic strategies based on the stages of development from the parasites.

## Figures and Tables

**Figure 1 insects-10-00130-f001:**
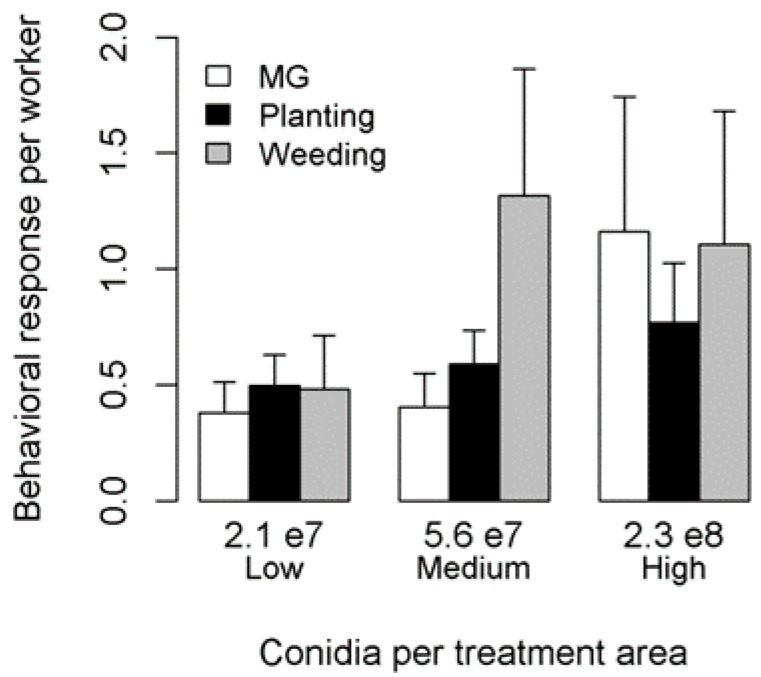
Ant workers’ behavioral response when exposed to three different *Metarhizium brunneum* conidia amounts. Error bars represent standard errors.

**Figure 2 insects-10-00130-f002:**
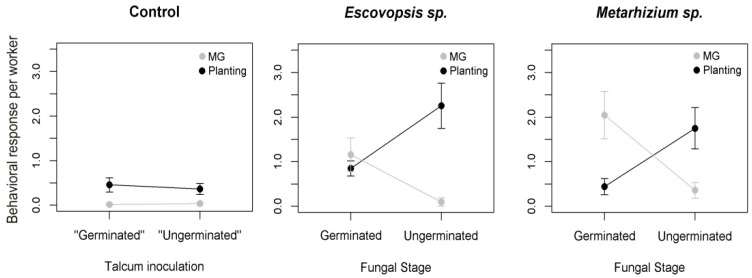
Mean ± standard errors frequencies of the hygienic behaviors metapleural gland (MG) grooming (MG: Grey dots) and fungal mutualist planting (black dots) exhibited by *Trachymyrmex* sp. 10 fungus-growing ants, in response to the germinated conidia or ungerminated conidia of the *Escovopis* and *Metarhizium* fungal parasites. Control treatments were talcum powder (for ungerminated conidia) or talcum powder contaminated by a fungal mutualist (for germinated conidia).

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
