# Peer review of "Hygiene Defense Behaviors Used by a Fungus-Growing Ant Depend on the Fungal Pathogen Stages"

_insects, 2019, doi:10.3390/insects10050130_

Round 1
Reviewer 1 Report
In this manuscript the authors investigate whether hygienic behaviors of the ant Trachymyrmex sp. 10 are dependent on parasite identity, stage or concentration. They found that hygienic behaviors are more dependent on stage than concentration. While I like the idea of the manuscript, the authors failed to report more details about the methods used (see below). Moreover, as concentration was tested only for one parasite, the authors should refrain from generalizing to both pathogens (see below).
Throughout the methods the authors report that they collected and used several ant colonies. However, at no point actual numbers are reported. This needs to be remedied and reported together with a clear statement of how many replicates/experimental groups per colony were used for each experiment.
I also miss a dedicated statistical analysis sub-header. While the authors say in line 82 that both experiments where analyzed with a PERMANOVA, the result section contains correlations, e.g. 88-89 and Permutation tests, e.g. line 93. In addition, it is not clear what was quantified (line 59). Is it time ants spend performing the different behaviors or maybe the proportion of ants performing the behaviors? This needs in my view more explanation.
Finally, although a PERMAVOA allows analysis of all behaviors together, and the PERMANOVA did show a non-significant difference according to Metarhizium concentration (line 86-87), there is also a clear indication from Figure 1 that the behavioral response is stronger with a high dose compared to a low dose for all behaviors. This should not be completely ignored and should at least be mentioned in the results or discussion.
Minor points:
Line 66-72: It is unclear to me why hygienic responses to different pathogen concentrations were only investigated for Metarhizium brunneum but not for Escovopsis. Please explain. Also, please change general statements regarding results on pathogen concentration, for example in line 21-23, in a way to make it clear that this applies only to Metarhizium.
Line 72: It is not completely clear to me whether the 10 experimental groups for each pathogen concentration correspond to 20 animals per experimental group. Please specifiy.
Figure 1: A reference to the Figure 1 is missing completely in the main text. Moreover, in this figure but also Figure 2 it is not clear to me what is reported on the y-axis (time, proportion of animals, ….?).
Line 110: Please remove the second “presence” in the sentence.
Line 116-123: Please think about referencing to Davis et al. 2018 (https://doi.org/10.1038/s41598-018-32721-7) as it appears to fit very well.
Line 133: Please change the italicized “germination and growth is sensitive to” to normal letters.
Line 134: Please remove “permit”
Line 133-137: Consider a more thorough explanation. I at least cannot follow the logic of these two sentences.
Line 137: Please consider changing “While, that using…” into “Using…”
Author Response
Response to Reviewer 1. We agree with the comments and recommendations of reviewer 1 to improve the manuscript and we have addressed all of them as follows (in blue).
Reviewer 1:
Comments and Suggestions for Authors
In this manuscript the authors investigate whether hygienic behaviors of the ant Trachymyrmex sp. 10 are dependent on parasite identity, stage or concentration. They found that hygienic behaviors are more dependent on stage than concentration. While I like the idea of the manuscript, the authors failed to report more details about the methods used (see below). Moreover, as concentration was tested only for one parasite, the authors should refrain from generalizing to both pathogens (see below).
Throughout the methods the authors report that they collected and used several ant colonies. However, at no point actual numbers are reported. This needs to be remedied and reported together with a clear statement of how many replicates/experimental groups per colony were used for each experiment.
R. We added the information:
Lines 81-84: “In this experiment, we used 30 colonies, and from each colony we formed one experimental group with 20 workers and 0.5 g of the fungus garden. We used 10 experimental group for each pathogen concentration. The colonies used for each concentration were selected randomly.”
Lines 93-95: “In this experiment, we used 60 colonies, and from each colony we formed 1 experimental group with 20 workers and 0.5 g of the fungus garden. We used 10 experimental group for each stages per pathogen and control. The colonies used for each concentration were selected randomly.”
I also miss a dedicated statistical analysis sub-header. While the authors say in line 82 that both experiments where analyzed with a PERMANOVA, the result section contains correlations, e.g. 88-89 and Permutation tests, e.g. line 93. In addition, it is not clear what was quantified (line 59). Is it time ants spend performing the different behaviors or maybe the proportion of ants performing the behaviors? This needs in my view more explanation.
R. We added the information, including sub-header:
Lines 100-109
“2.4 Statistical Analyses
For the analyzes, we recorded the total number of times that each behavior was observed during one hour, and this was divided by the average number of workers attending the fungus garden at the beginning of each 10 minute period during one hour (following methods used in [11]). The observations were conducted with the aid of a 70X stereomicroscope. Data from both experiments were analysed with PERMANOVA [16]. For the concentration experiment, we explored the possibility of collinearity between behavioral responses for the experiment with different pathogen concentrations by using Pearson correlations. For the stages experiment, in addition to PERMANOVA, we used one-way permutation tests as a post-hoc analysis for pairwise comparisons.”
Finally, although a PERMAVOA allows analysis of all behaviors together, and the PERMANOVA did show a non-significant difference according to Metarhizium concentration (line 86-87), there is also a clear indication from Figure 1 that the behavioral response is stronger with a high dose compared to a low dose for all behaviors. This should not be completely ignored and should at least be mentioned in the results or discussion.
R. We added:
Lines 145-149: “Moreover, ants did not adjust their behavior based on the conidia concentrations tested. However, our results show a tendency that increasing M. brunneum conidia concentration increased the hygiene behaviors. Previous studies reported a positive association between an increase on the pathogen concentration and use of the hygiene defense [11].”
Minor points:
Line 66-72: It is unclear to me why hygienic responses to different pathogen concentrations were only investigated for Metarhizium brunneum but not for Escovopsis. Please explain. Also, please change general statements regarding results on pathogen concentration, for example in line 21-23, in a way to make it clear that this applies only to Metarhizium.
R. We used only Metarhizium brunneum to conduct the concentration experiment because we lost some nests in the lab, and we did not have sufficient colonies to conduct experiments of concentration of Escovopsis.
However, to clarify we added the information as:
Lines 21-24 “By inoculating Trachymyrmex colonies with different parasite concentrations (Metarhizium) or stage (germinated conidia or ungerminated conidia of Metarhizium and Escovopsis),”
Line 72: It is not completely clear to me whether the 10 experimental groups for each pathogen concentration correspond to 20 animals per experimental group. Please specifiy.
R. Clarified in lines 82-85, and 93-96.
Figure 1: A reference to the Figure 1 is missing completely in the main text. Moreover, in this figure but also Figure 2 it is not clear to me what is reported on the y-axis (time, proportion of animals, ….?).
R. We added the reference of the figure 1 in line (114), and clarified the y-axis added to both figures, the information “per worker”.
Line 110: Please remove the second “presence” in the sentence.
R. Removed
Line 116-123: Please think about referencing to Davis et al. 2018 (https://doi.org/10.1038/s41598-018-32721-7) as it appears to fit very well.
R. Reference has been added in the text. Lines 166-167 and lines 246-248.
Line 133: Please change the italicized “germination and growth is sensitive to” to normal letters.
R. Changed.
Line 134: Please remove “permit”
R. Removed.
Line 133-137: Consider a more thorough explanation. I at least cannot follow the logic of these two sentences.
R. These sentences are required to explain the costs of both strategies to the ants. Any more comments are speculative, given our understanding.
Line 137: Please consider changing “While, that using…” into “Using…”
R. Changed.
We hope that our changes have sufficiently improved the manuscript and that it will be of interest for publication in Insects. Thank you for your consideration of the revision and we look forward to your decision.
With best regards,
Hermógenes Fernández-Marín
Reviewer 2 Report
Minor edits in the attached PDF. Overall, nice paper, well designed study, and interesting results.

Author Response
Dear Assigned Editor Barbara Wang:
Thank you for sending on the comments by the referees. We agree with their comments and edits, which have improved the manuscript. We respond specifically to each point suggested by the reviewers, below.
We have addressed all of the reviewers’ comments and hope that you find that the manuscript is satisfactory for publication.
Responses to Reviewer comments:
Respond to Reviewer 2.
We agree with the comments and recommendations of reviewer 2 to improve the manuscript and we have addressed all of them as follows (in blue)
R. Line 6-13: Affiliates format were unified.
R. Lines 42, 58, 59: authories were added.
R. Line 131: word size unified.
We hope that our changes have sufficiently improved the manuscript and that it will be of interest for publication in Insects. Thank you for your consideration of the revision and we look forward to your decision.
With best regards,
Hermógenes Fernández-Marín
Round 2
Reviewer 1 Report
In the revised version the authors have satisfactorily addressed all major points raised on the previous version of the manuscript.
However some changes need little corrections:
Line 78 and 90: "We used 10 experimental group..." to "We used 10 experimental groups..."
Line 151: "...immuny system." to "...immune system."
And I still think that the added sentence (line 160-161) together with the sentences (now 161-165, previously 133-137) are hard to follow in their logik, but this might be just me.